# Mitochondrial Impairment Induced by Sub-Chronic Exposure to Multi-Walled Carbon Nanotubes

**DOI:** 10.3390/ijerph16050792

**Published:** 2019-03-05

**Authors:** Giuseppa Visalli, Alessio Facciolà, Monica Currò, Pasqualina Laganà, Vincenza La Fauci, Daniela Iannazzo, Alessandro Pistone, Angela Di Pietro

**Affiliations:** 1Department of Biomedical and Dental Sciences and Morphofunctional Imaging. University of Messina, 98125 Messina, Italy; gvisalli@unime.it (G.V.); moncurro@unime.it (M.C.); plagana@unime.it (P.L.); vlafauci@unime.it (V.L.F.); 2Department of Clinical and Experimental Medicine, Unit of Infectious Diseases, University of Messina, 98125 Messina, Italy; afacciola@unime.it; 3Department of Electronic Engineering, Industrial Chemistry and Engineering, University of Messina, 98125 Messina, Italy; diannazzo@unime.it (D.I.); pistone@unime.it (A.P.)

**Keywords:** MWCNTs, oxidative stress, mitochondria

## Abstract

Human exposure to carbon nanotubes (CNTs) can cause health issues due to their chemical-physical features and biological interactions. These nanostructures cause oxidative stress, also due to endogenous reactive oxygen species (ROS) production, which increases following mitochondrial impairment. The aim of this in vitro study was to assess the health effects, due to mitochondrial dysfunction, caused by a sub-chronic exposure to a non-acutely toxic dose of multi walled CNTs (raw and functionalised). The A549 cells were exposed to multi-walled carbon nanotubes (MWCNTs) (2 µg mL^−1^) for 36 days. Periodically, cellular dehydrogenases, pyruvate dehydrogenase kinase 1 (PDK1), cytochrome c release, permeability transition pore (mPTP) opening, transmembrane potential (Δψ m), apoptotic cells, and intracellular ROS were measured. The results, compared to untreated cells and to positive control formed by cells treated with MWCNTs (20 µg mL^−1^), highlighted the efficiency of homeostasis to counteract ROS overproduction, but a *restitutio ad integrum* of mitochondrial functionality was not observed. Despite the tendency to restore, the mitochondrial impairment persisted. Overall, the results underlined the tissue damage that can arise following sub-chronic exposure to MWCNTs.

## 1. Introduction

With the growing trend in the production and applications of carbon nanotubes (CNTs), the increasing use in composite materials [1] and their exploration as innovative solutions for biomedical applications [2,3,4], there will be a corresponding increase in potential human exposures [5]. Such exposures can cause substantial health issues as a result of the chemical–physical features and biological interactions of CNTs. Numerous efforts have been made over the past two decades to investigate the biocompatibility and toxicological effects of CNTs, which are still poorly understood and controversial [6,7,8,9].

The harmful effects of CNTs have been shown to be highly dependent on the type of cells used for in vitro studies [10] and on the heterogeneity of the produced CNTs. The length, diameter, structural defects, surface area, tendency to agglomerate dispersibility in water solution, presence and nature of metal catalyst residues [11,12] and surface chemistry [13,14,15,16] greatly influence the biological reactivity of CNTs.

Oxidative stress is a common mechanism involved in the cytotoxicity of these nanostructures [17]; following inhalation, the CNTs penetrate deeply in the respiratory tract and cause a strong pro-oxidant effect. This effect, which could trigger carcinogenic asbestos-like mechanisms, was previously observed by our research group in an in vitro cell model of alveolar epithelium and neuronal-like cells after short-term exposures to multi-walled CNTs synthesized in our laboratory [18,19]. Consistent with an overproduction of reactive oxygen species (ROS), we detected significant time- and dose-dependent increases in lipid peroxidation and mitochondrial impairment.

Polyunsaturated fatty acids (PUFAs), present in the mitochondrial membrane, are particularly susceptible to free radical-initiated oxidation, which determines the decrease in the mitochondrial transmembrane potential and metabolic impairment because of the shift from oxidative phosphorylation to anaerobic glycolysis. Mitochondrial damage, by releasing caspase-activating proteins, can trigger the intrinsic apoptotic pathway, as observed in the alveolar cell line exposed to airborne particles [20,21], asbestos [22], metals and oil fly ash [23,24,25].

Mitochondria are the powerhouses of cells and, at the same time, the suicidal weapon store because they are the primary sources of intracellular ROS production. It has been estimated that physiologically, during oxidative phosphorylation, 2–4% of the oxygen is converted to O_2_^−^ and, then, to H_2_O_2_. Dozens of lethal signal transduction pathways converge on mitochondria and cause the permeabilisation of the mitochondrial outer membrane, leading to the cytosolic release of pro-apoptotic proteins and to the impairment of the bioenergetic functions of mitochondria [26].

The increased residence time of electrons in complexes I and III of the respiratory chain [27] causes a depolarisation of the inner mitochondrial membrane (∆ψm), which can further amplify the redox imbalance MWCNT-induced through the production of endogenous ROS. Several studies confirmed the destruction of mitochondrial membrane potential (∆ψm) and mitochondrial swelling by CNTs; both events were produced as a result of the increased formation of ROS, the release of cytochrome c (cyt c), the disturbance of the mitochondrial electron transfer chain (mtETC) complexes and the collapse of the mitochondria [6,28,29].

ROS are determinants for the release of the pro-apoptotic cyt c because cyt c is bound to the inner membrane by anionic phospholipids, specifically cardiolipin (CL), that are highly susceptible to peroxidation [30]. The proteins, Bax and Bak, increasing mitochondrial outer membrane permeabilisation, promote cyt c release and apoptosis [31,32].

Mitochondrial permeability transition pore (mPTP) is another key participant in mitochondrial apoptosis leading to the mitochondrial depletion of Ca^2+^ and to the release of cyt c that trigger caspase activation [33].

Most of the in vivo and in vitro studies reported in the scientific literature [3,6,10] were performed to evaluate acute effects of CNTs using higher doses compared to the doses at which the population may be exposed. Conversely, the aim of the present study was to increase knowledge on the health effects, focusing on the effects on mitochondria function, of sub-chronic exposure to a non-acutely toxic dose, which is the most realistic dose, especially for people occupationally exposed.

## 2. Materials and Methods

### 2.1. Pristine and Functionalized MWCNTs

We examined homemade raw (i.e., pristine) MWCNTs (which are called pMWCNT) and functionalised MWCNTs (that is, MWCNT-COOH, named fMWCNTs); these two MWCNTs have been widely studied for the assessment of short-term toxicity in two different cell lines [18,19]. pMWCNTs were synthesised by catalytic chemical vapour deposition (CCVD) and subsequently purified from both free metals and carbonaceous particles, by products of the synthesis process [34,35]. The covalent insertion of a carboxylic group was obtained by strong acidic oxidation. Both MWCNTs were carefully characterised by thermogravimetric analysis (TGA), UV (ultraviolet) spectra, scanning electron microscopy and high-resolution transmission electron microscopy, as previously reported [18]. In addition, abiotic and in vitro experiments were performed to assess the bioavailability of iron that was used as a catalyst in the MWCNT synthesis. Iron was detected by atomic absorption spectroscopy analysis and was equal to 2.5–2.8%. This was almost fully comprised of Fe_2_O_3_ and was not bioavailable [12].

Due to the strong hydrophobicity, common to all CNTs, and to the van der Waals forces occurring at the surface, the concentrated MWCNT suspensions (100 × in PBS (Phosphate buffered saline)) were sonicated for 20 min in an ice bath (frequency 40 kHz). Moreover, just before all in vitro experiments, the MWCNT work suspensions were, further, sonicated for 3 min in the culture medium containing 10% fetal bovine serum (FBS) to enhance their dispersibility by the protein content of the cell medium.

### 2.2. Cell Cultures and Exposure Conditions

The lung alveolar region is the prime site of deposition for inhaled particles, including engineered nanomaterials, which, due to their size, fall within the breathable fraction. Therefore, we used the human alveolar cell line A549 as a pulmonary-like cell system. Cells were cultured in RPMI medium with 2 mM L-glutamine, 10% (v/v) FBS, 100 IU mL^−1^ penicillin and 100 µg mL^−1^ streptomycin at 37 °C in a humidified 5% CO_2_ atmosphere.

Based on data previously obtained using the same cell model and from the literature regarding possible exposures in the workplace [6,17,18], we assessed the effects of sub-chronic exposure to 2 µg mL^−1^ of CNTs suspensions. In parallel, cell monolayers were exposed to pMWCNTs and fMWCNTs. The cells were maintained in the same medium, with the addition of PBS instead of the CNT suspensions, to serve as a negative control, and cells treated with MWCNT suspensions at 20 µg mL^−1^ were used as a positive control. The 20 µg mL^−1^ concentration was chosen to guarantee a sufficient level of survival of the exposed cells for the times established by the experimental protocol. Every three days, cell medium with and without MWCNTs was changed and treated and non-treated monolayers were sub-cultured weekly. At 1, 7, 21, 28 and 36 days, aliquots were sampled to perform the analyses. In parallel, cell viability was evaluated by the trypan blue exclusion test using a 0.4% dye solution. To confirm MWCNTs–cell interaction, previously observed in short time examinations, we carried out a qualitative analysis, and A549 semiconfluent monolayers were observed by transmission microscopy.

### 2.3. Evaluation of Mitochondrial and Cellular Enzymatic Activity and Cytochrome c Release

To assess the mitochondrial function, we used western blot analysis to detect the expression of pyruvate dehydrogenase kinase 1 (PDK1) and the release of cytochrome c in the cytosol fraction devoid of mitochondria.

To measure PDK1 protein expression, cells were lysed for 10 min on ice with RIPA buffer (1% NP-40 or Triton X-100, 1% sodium deoxycholate, 0.1% SDS (Sodium Dodecyl Sulphate), 150 mM NaCl, 50 mM Tris-HCl, pH 7.8, 1 mM EDTA (Ethylenediaminetetraacetic acid)) supplemented with a protease inhibitor cocktail. Moreover, to evaluate the mitochondrial release of cytochrome we resuspended the cells in 200 μL of STM buffer (250 mM sucrose, 50 mM Tris-HCl pH 7.4, 5 mM MgCl_2_, protease and phosphatase inhibitor cocktails) and homogenized for 1 min on ice using a tight-fitting Teflon pestle. The homogenate was decanted into a centrifuge tube and maintained on ice for 30 min, vortexed at maximum speed for 15 s, and then centrifuged at 800× *g* for 15 min to obtain cytosolic and mitochondrial fractions in the supernatant. Then, by a further centrifugation of the supernatant at 11,000× *g* for 10 min, a cytosol fraction devoid of mitochondria was obtained, and 30 µg of this latter supernatant was fractionated on SDS-PAGE. Subsequently, they were electrically transferred to a nitrocellulose membrane (Millipore, Rodano, Italy) and were blocked with 5% non-fat dry milk in TBS-T buffer (10 mM Tris-base, 10 mM NaCl, and 0.1% Tween-20) overnight at 4 °C. Then the membranes were probed with mouse anti-PDK1 monoclonal antibody (diluted 1:500 in TBS-T), anti-cytochrome c (diluted 1:100 in TBS-T), and β-actin (diluted 1:3.000 in TBS-T) for 2 h at room temperature followed by incubation with horseradish peroxidase-conjugated anti-mouse secondary antibodies (respectively diluted 1:1.500, 1:1.000 and 1:10.000 in TBS-T) (Sigma-Aldrich, Milan, Italy). Immunoblots were developed with an ECL kit on Kodak film. After normalization against β-actin, blots were scanned and quantified by densitometric analysis with Image J 1.47 (http://imagej.nih.gov/ij/).

Moreover, cellular dehydrogenases, including succinate dehydrogenases (SDH in mitochondrial complex II), were detected by measuring the reduction of 3-(4,5-dimethylthiazol-2yl)-2,5-diphenyltetrazolium bromide (MTT). Analyses were performed on aliquots of each sample. Briefly, cell suspensions were normalised to a final concentration of 1 × 10^5^ mL^−1^, transferred to microplates (100 µL/well), and 0.4% of MTT was added before incubation at 37 °C for 3 h. Then, a mixture composed of 50 mM HEPES (*4-(2-hydroxyethyl)-1-piperazineethanesulfonic acid*), pH 8.0, and ethanol (1:9, v/v) was added to solubilise the resulting violet coloured formazan crystals. Absorbance of the dye was measured at 540 nm using a microtiter plate reader (Tecan Italia, Milan, Italy). The enzymatic activity was obtained by calculating the product between the absorbance values and the dilution factors. The values obtained were compared with the negative control (100%).

### 2.4. Assessment of mPTP Opening

Mitochondrial permeability transition pore (mPTP) opening was evaluated by using the calcein-AM/cobalt method [36]. After treatment with MWCNTs, the cell aliquots, prepared as described above, were seeded in 96-well plates (100 µL/well) and loaded with 5 µM calcein-AM and 0.5 mM CoCl_2_ (cytosolic calcein quencher) in PBS for 15 min at 37 °C. The cells were analysed by a microplate reader with an excitation wavelength of 488 nm and an emission wavelength of 525 nm. Decreases in fluorescence were indicative of the loss of calcein due to the mPTP opening.

### 2.5. Fluorimetric Analysis for the Assessment of Mitochondrial Transmembrane Potential and ROS Production

The effect of MWCNT–cell interactions on mitochondrial function was further evaluated by analysing mitochondrial transmembrane potential. This was accomplished by measuring the incorporation of the fluorescent probe rhodamine 123 (R123, 10 µM) (Invitrogen Molecular Probes, Milan, Italy). Based on its chemical properties, this cationic fluorochrome crosses the mitochondrial membrane and it is stored in the matrix of functional mitochondria with a transmembrane potential (Δψ_m_) that is indicative of an active proton gradient during oxidative phosphorylation.

Intracellular ROS were detected using the fluorochrome 2′,7′-dichlorofluorescein-diacetate (DCF-DA) (1 µM). For the analyses, treated cells were washed three times with PBS containing 10 mM D-glucose at pH 7.4. Then, aliquots of cell suspensions (1 × 10^5^ mL^−1^) were prepared in the same buffer and the emission values were read both before and after the separately addition of the respective probe, in order to subtract emission values that may have been due to auto-florescence. The probe-loaded cells were incubated for the times indicated and at the temperatures established in the study protocol. Fluorescence intensity was measured using a microplate reader (Tecan Italia), and the wavelengths of excitation and emission were 535–595 for R123 and 485–535 for DCF-DA, respectively. For each analysis, the obtained values were used to calculate the percentage changes (%Δ) compared to untreated cells.

### 2.6. Apoptosis Detection

Apoptotic cells were assessed by fluorimetric detection using AnnexinV-FITC (Sigma-Aldrich, Milan, Italy). The analysis is based on the changes of phosphatidylserine (PS) position in the cell membrane and in its outsourcing in apoptotic cells. Cell suspensions were obtained by combining the cells that were recovered using 0.25% trypsin and 1 mM EDTA with the ones suspended in medium. Briefly, cells from all subcultures were harvested by centrifugation at 1000× *g* for 5 min and were resuspended (approximately 1 × 10^6^ cells mL^−1^) in Annexin Binding Buffer 1× (100 mM HEPES/NaOH, pH 7.5, 1.4 M NaCl and 25 mM CaCl_2_) containing Annexin V-FITC (0.5 μg mL^−1^). After 20 min of incubation at 37 °C, cell suspensions were centrifugated washed twice and resuspended in 100 μL, using the same buffer, and then transferred into 96 well microplates. In a microplate reader (Tecan Italia) and by using 485 nm as wavelength of excitation and 535 nm for emission, the emission values were measured to calculate the percentage changes (%Δ) of apoptotic cells compared to untreated cells.

### 2.7. Cell Proliferation Index

An overview of the effects of MWCNT was obtained by determining the cellular proliferation index. Starting from the same number of cells for each treatment, the index was obtained by the cell count that was periodically carried out on the sample aliquots. The average values at the intervals that were assayed were compared to the values recorded in the control cells.

### 2.8. Statistical Analyses

All data are presented as mean ± the standard error of the mean (SEM) based on at least three independent experiments. Data were analysed by one-way analysis of variance (ANOVA), and multiple comparisons of the means were performed by the Tukey–Kramer test (GraphPAD Software for Science, San Diego, CA, USA). The relationships between different parameters were assessed by the Pearson correlation coefficient. Significance was accepted at *p* < 0.05.

## 3. Results

### 3.1. MWCNT Effects in Lung Epithelial Cells

To evaluate the effects of sub-chronic exposures at occupationally realistic doses of p- and f-MWCNTs, we preliminarily checked the interactions of these two MWCNTs with lung epithelial cells. Our qualitative assessment confirmed what was obtained previously in the same acutely exposed cell model, and it highlighted the concentration and time-dependent effects of MWCNTs. As shown by the microscopic observation (Figure 1), the presence of dark aggregates in or on the cells was clearly visible in A549 monolayers treated with both types of MWCNTs at the highest dose (positive controls). Moreover, at this dose, some spaces were observed, indicating the detachment of dead cells that were more numerous in pMWCNTs-treated cells. Instead, at 2 μg mL^−1^, the effects of the nanotube in the monolayers were barely detectable, especially in f-MWCNTs- treated cells, and the cellular morphology was superimposable to that of the control cells. In these experiments, cell viability was assessed measuring the percentage of dead cells. 

The table in Figure 1 reports the results obtained by the trypan blue exclusion test. Compared to untreated cells, significantly higher percentages of dead cells were observed after 1 day in positive controls (20 µg mL^−1^) for both p- and f-MWCNTs (*p* < 0.01). Despite the fact that the number of dead cells was almost halved after 36 days of exposure, the differences in comparison to control cells were maintained and were very significant (*p* < 0.01). Even at the tested dose, the percentage of dead cells in the monolayers treated for 1 day were significantly higher (*p* < 0.05). However, treatment for 36 days decreased these percentages that, in comparison to untreated cells, were not significant. The trend of cellular mortality showed an adaptability inversely related to MWCNT sub-chronic exposure dose in the sub-cultured cells.

### 3.2. Mitochondrial Impairment and MWCNT-Induced Apoptosis

A wide spectrum of analyses was performed to assess the mitochondrial compartment and the MWCN-induced apoptosis. In particular, the expression of PDK1, the release of cytochrome c, the transmembrane potential, the mPTP opening, and the outsourcing of phosphatidylserine were evaluated. The MWCNTs-induced inhibition of mitochondrial activity was detected by assessing the expression of PDK1 that regulates glucose metabolism by phosphorylation of the E1α subunit of mitochondrial pyruvate dehydrogenases. Figure 2A shows a representative result of a Western blot analysis that highlights the dose dependent higher levels of PDK1, in MWCNTs-treated cells in the short time. Despite the fact that PDK1 levels decreased over time, limiting the effects on TCA cycle a full metabolic recovery was never completed in the examined time interval. Furthermore, the Western blot results of cytochrome c in the cytosolic fraction devoid of mitochondria are reported in Figure 2A. The higher levels of cytochrome c, observed after short exposure, confirmed the MWCNTs-induced mitochondrial impairment that was partially counteracted over time. Mitochondrial impairment was also assayed by measuring the transmembrane potential, which is strongly related to the active proton gradient that is maintained during oxidative phosphorylation in metabolically active cells. The measurement of R123 emission underlined the MWCNTs-induced mitochondrial dysfunction, and Δψ_m_ values were almost halved at 1 and 7 days (*p* < 0.05) in positive controls of both nanotube types, highlighting the organelle collapse (Figure 2B). Although not significant, even at the occupationally realistic dose tested, Δψ_m_ decreased and the values were within 65% of the control cell values. Analogously to what was observed with the analysis of the enzymatic activity, the transmembrane potential also increased progressively over time, confirming a tendential but not complete recovery of mitochondrial function in cell progeny that were steadily exposed to MWCNTs. In fact, compared to control cells, Δψ_m_ values in cells treated for 36 days were 75 and 83% for p- and f-MWCNTs, respectively.

The results obtained by mPTP opening further confirmed MWCNTs-induced mitochondrial impairment (Figure 2C). The emission values of calcein were decreased in cells treated for 1d, underling mitochondrial release of calcein in the citosol, where its fluorescence was quenched by the Co which is unable to cross mitochondrial membrane. However, in sub cultured cells, a recovery was observed and, at the end of the examined interval, the emission values recorded in cells treated with 2 μg mL^−1^ were 10% and 5% below the control cells for cells treated with p- and f-MWCNTs respectively. As expected, the results of apoptotic cells obtained by using the outsourcing of phosphatidylserine, as a marker of apoptosis, were strongly related to all assayed mitochondrial parameters. Figure 2D reports that the %Δ of apoptotic cells compared to untreated cells was higher after exposure for 1d to both types of MWCNTs and for both doses. In the following subcultures, apoptotic cells decreased over time, especially in cells treated at the tested dose of MWCNTs. This confirms the adaptability of alveolar cells to the exposure conditions. However, they were not totally restored to baseline conditions. The results of the Pearson test showed highly significant correlations between apoptotic cells and cytochrome c release, Δψ_m_, and mPTP opening, respectively (*p* < 0.01).

### 3.3. Cellular Enzymatic Activity and ROS Production in Exposed Cells

To assess cellular dehydrogenases, included succinate dehydrogenases (SDH in mitochondrial complex II), the time course of MTT reduction was measured in cells exposed to MWCNTs (Figure 3A). In the positive controls exposed for 1 day to both MWCNTs, the activity of cellular dehydrogenases was 60% below (*p* < 0.01) that of untreated cells. Instead, the absorbance values in cells exposed for 1 day at 2 μg mL^−1^ was not significantly different from the negative control, and dehydrogenase activity was equal to 67.1% and 74.5% for pristine and oxidised nanotubes, respectively. Over time, we observed a progressive increase of the absorbance values, highlighting an adaptation of the sub cultured cells to the steady exposure. Despite the increased absorbance values at 36 days, which was higher in the f-MWCNTs-treated cells, the activity of cellular dehydrogenases did not overlap the control cells. This emphasizes that MWCNT exposure, even at a low dose, caused cellular damage, albeit of a limited extent. Considering that the MTT values were strongly and inversely related to apoptotic cells and to Trypan blue values (*p* < 0.01), it is highly plausible to believe that the reduction of dehydrogenase activity was not due to a mere enzymatic inhibition, but rather to a reduction in the number of viable cells.

Figure 3B reports the ROS levels in exposed cells compared to control cells. As previously detected in the same cell model, a stronger pro-oxidant effect was caused by pMWCNTs than fMWCNTs, confirming what was observed in the short-term exposures to high doses. In comparison to positive controls after 7 and 21 days, the A549 cells exposed at 2 µg mL^−1^ were able to almost completely counteract the oxidative stress induced by f-and p-MWCNTs, respectively.

Conversely, ROS production remained higher in the positive control sample (20 µg mL^−1^) of pristine nanotubes than in untreated cells, and the observed values were on average 25% higher than the control after 36 days.

### 3.4. Proliferation Index

The cellular proliferation index was used for an overview of the MWCNT-cell interactions. In comparison to untreated cells, whose average values of proliferation index in the interval assayed was arbitrarily set at 100; the values at 2 µg mL^−1^ were 90.9 (±5.4) and 93.8 (±6.2) for p-and f-MWCNTs, respectively. The values in positive controls were further decreased and equal to 75.1 (±6.5) and 78.4 (±5.9), underlining the harmful effects of the engineered nanoparticles.

## 4. Discussion

The production of safer nanofibers is extremely important considering the ease with which lightweight nano-sized MWCNTs aerosolize, increasing the risk of developing pulmonary disorders after their inhalation [3,8,15]. Once inhaled, the high hydrophobicity of the CNTs promotes their interaction with the plasma membrane surface. The internalisation pathways of nanotubes include the energy dependent endocytosis (i.e., pinocytosis or, only in specialised cells, phagocytosis) and passive diffusion. Due to the high length to diameter ratio, the latter pathway allows MWCNTs to efficiently cross the phospholipid bilayer of cell membranes. As established by several studies, both mechanisms do not alter membrane integrity [37,38,39,40]. Unlike the endocytosis-mediated internalisation of CNT tangles, passive diffusion reduces cytotoxicity because it does not cause lysosomial content leakage by overloading endosomes [12]. However, regardless of the internalization process, nanotubes cause oxidative damage in cell compartments.

The results were obtained by simulating sub-chronic exposure in a work environment and they highlighted the efficiency of homeostatic mechanisms to counteract over time MWCNTs-induced ROS overproduction. However, the mitochondria were not entirely exempt from the harmful effects of the engineered nanoparticles, which caused apoptosis in a portion of exposed cells. Mitochondria are dynamic bioenergetic semiautonomous organelles that execute a myriad of functions pertaining to cellular metabolism and homoeostasis. In addition to cellular energy generation via oxidative phosphorylation (OXPHOS), they play a central role in calcium homoeostasis, initiation of caspase-dependent apoptosis, cellular stress response, sulphur metabolism, and biosynthetic processes [41,42,43,44,45,46,47]. The paramount importance of mitochondrial integrity for the proper functioning of the OXPHOS system is well known. It comprises five multimeric enzymes (complexes I to V) that are incorporated into super-complexes to reduce ROS levels and two mobile electron carriers (coenzyme Q 10 and cyt c) [48]. Despite the clear tendency to restore full functionality, our results showed that the MWCNTs-induced mitochondrial impairment persisted over time not only in the positive controls but also at the dose to which workers could conceivably be exposed. Considering the strong association between depolarization and electron transport impairment, this effect was highlighted by the transmembrane potential values.

The superimposable effects of both CNTs could be imputable to excessive lengths with regard to the p-MWCNTs (10–20 μm vs. 200–1000 nm) and to higher surface reactivity with regard to the f-MWCNTs [12,19]. Even if the presence of carboxyl groups enhances water dispersibility and causes a reduction in the length to diameter ratio of CNTs, making them more biocompatible, these effects are nullified by the acid-induced erosion in the graphene external layers. This increases the surface reactivity and, consequently, cellular toxicity [18].

Mitochondria are signalling organelles that constantly regulate the production of energy according to the needs of the cell. At the same time, the mitochondria are able to manage cell behaviour by retrograde signalling to other cell compartments. These signals include mitochondrial ROS, transmembrane potential and calcium fluxes across the mitochondrial membrane as well as AMP/ATP and NAD^+^/NADH ratios as bioenergetic and redox signals, respectively. While mitochondrial ROS (mtROS) overproduction and/or hiROS (high ROS, such as hydroxyl radicals or peroxynitrites) are liable for cellular pathology by directly damaging biomolecules, physiological levels of mitochondrial loROS (low ROS, such as superoxide or hydrogen peroxide) modulate gene expression and cell survival. Despite the positive recovery results observed in the present study, sub-chronic exposures to our homemade MWCNTs did not allow a *restitutio ad integrum* of mitochondrial functionality. Considering the functions of the mitochondria as summarised briefly above, the values of Δψ_m_, equal to 75% and 83% for p- and f-MWCNTS, suggested a metabolic impairment in the exposed cells. Those findings were confirmed by the results of cytochrome c, which highlighted the way in which mitochondrial dysfunction unavoidably affected cells by triggering a mitochondrial apoptotic pathway. The dissociation of cyt c from the inner mitochondrial membrane, followed by the release of cyt c from the intermembrane space to the cytosol through the outer membrane, is regulated by a mechanism involving two pro-apoptotic Bcl-2 family proteins, Bax and Bak, which are able to increase mitochondrial outer membrane permeabilization [31,32]. This pathway was clearly underlined by the release of cyt c that was recorded in the cytosolic fraction devoid of the mitochondria of MWCNTs–treated cells and, downstream, by the increase of apoptotic cells as shown by the outsourcing of phosphatidylserine, which is a marker of apoptosis.

The pathogenic mechanism was further supported by the results of the calcein test performed to assess mPTP opening. mPTP is a key participant in mitochondrial apoptosis that has catastrophic consequences on the fate of cells, leading to the release of Ca^2+^ from the mitochondrial matrix and causing energetic collapse by depletion of adenosine triphosphate (ATP). Furthermore, mPTP, formed by voltage-dependent anion channel (VDAC), adenine nucleotide translocator 1 (ANT1), and cyclophilin-D (CypD), a peptidyl–prolyl–cis, trans-isomerase [48], causes the release of cytochrome c that triggers an apoptotic pathway via caspases activation. During stress conditions, such as exposure to engineered nanoparticles, the bond in the inner mitochondrial membrane (IMM) between CypD and ANT1 causes mPTP opening [49]. Even if we observed a partial recovery over time, the MWCNTs-induced mitochondrial impairment persisted and prevented the restoration of physiological conditions.

Several mitochondrial quality control mechanisms (QC) form a hierarchical network of interacting pathways that act at the molecular, organelle, and cell levels to maintain a “healthy” mitochondrial population in each cell. At the molecular level, ROS scavenging detoxifies superoxide and H_2_O_2_ through nuclear encoded Mn superoxide dismutase (MnSod) and glutathione peroxidase or peroxiredoxin. Considering the dual role of ROS reported above, a fine-tuned regulation of cellular ROS level is required in order to hinder damage (by abstraction of electrons) to proteins, lipids, and nucleic acids when threshold levels are overwhelmed. In the integrate network of mitochondrial quality control, this first protective line formed by ROS scavenging would seem sufficiently efficient after sub-chronic exposure to MWCNTs. The decreased levels of ROS suggest that the enzymes and the small molecules required in the scavenging process, such as glutathione and thioredoxin, were effectively regulated, counteracting the over production of ROS that boosts the pro-oxidant effect of engineered NPs.

Conversely, our data show how the MWCNTs-induced damage exceed the threshold within which downstream mechanisms of mitochondrial QC are able to maintain only functioning mitochondria. These QC mechanisms include repair of damaged components to restore their function and/or mitophagy (i.e., autophagy of whole dysfunctional organelles), and they control the balance (i.e., mitohormesis) between mitochondrial fusion and fission from which, thanks to biogenesis, a fully functional population of mitochondria can be generated, ensuring cell survival [50,51,52]. As suggested by the mitochondrial depolarization, the impairment of enzymatic activity and the increased mPTP opening, sub-chronic exposure to our highly purified MWCNTs causes a saturation of mitochondrial quality control mechanisms in alveolar cells. The presence of dysfunctional mitochondria affected the cellular proliferation index underling the potential pathogenic effects of MWCNTs-cell interaction in the onset of chronic obstructive pulmonary disease in the occupationally exposed subjects.

These results, obtained by simulating sub-chronic exposure to MWCNTs in a work environment, reduce the alarming picture that has emerged from many short-term studies that used unrealistic CNTs concentrations. Over time, ROS overproduction in alveolar epithelial cells was minimized due to the excellent mechanisms of homeostasis. However, even if the oxidative stress induced by the MWCNTs tested dose was temporary, the mitochondria were not entirely exempt from the harmful effects of the engineered nanoparticles. Considering the key role of mitochondria in managing the vital functions of cells, it is useful to underline the tissue damage that can arise following exposure to these synthetic nanoparticles.

## Figures and Tables

**Figure 1 ijerph-16-00792-f001:**
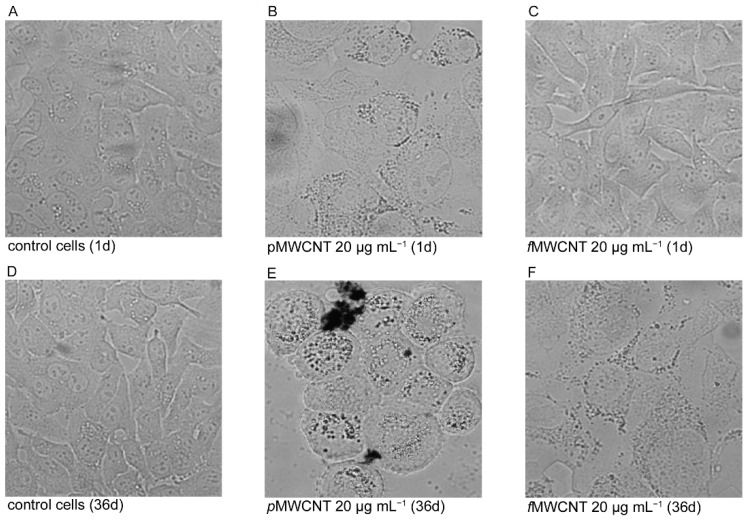
Representative phase contrast microscopy images of A549 semiconfluent monolayers to assess MWCNT-cell interactions. (**A**–**C**) control cells and cells exposed for 1 day to p- and fMWCNTs (20 µg mL^−1^). (**D**–**F**) control cells and cells exposed for 36 days to p- and fMWCNTs (20 µg mL^−1^). The images of cells treated with 2 μg mL^−1^ are not shown because the interactions were barely detectable and the cellular morphology was superimposable to that of the control cells. Table reports the percentages of dead cells, as assessed by the trypan blue exclusion test.

**Figure 2 ijerph-16-00792-f002:**
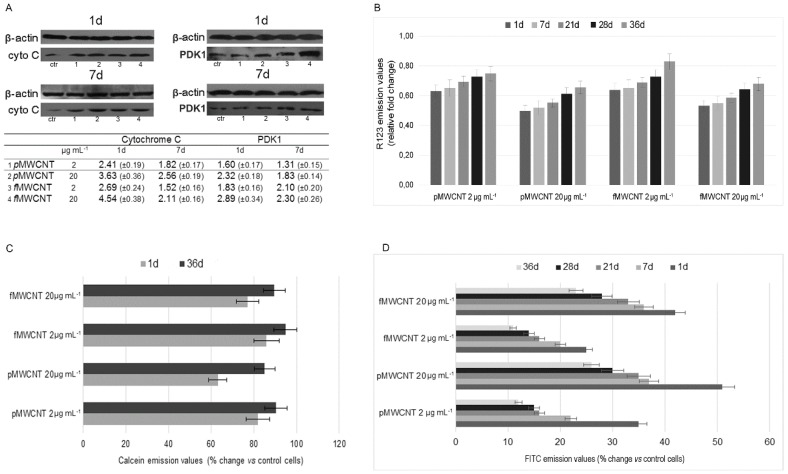
Assessment of mitochondrial function and apoptosis on MWCNT treated cells over time. (**A**) Representative result of western blot analysis performed to evaluate the expression of PDK 1 and to assess cyt c release (ctr: control cells). In the table, the results are expressed as relative fold changes. (**B**) Time course of R123 emission values that were used to assess the Δψ m in A549 cells. The results are expressed as relative fold change in comparison to control cells. (**C**) Results of the calcein-AM/cobalt method that was used to assess MWCNTs-induced mPTP opening during the examined interval. (**D**) Assessment of MWCNT-induced apoptosis by analysis of the outsourcing of phosphatidylserine over time. The results are expressed as percent change in comparison to control cells. All data are presented as mean ± SEM based on at least three independent experiments.

**Figure 3 ijerph-16-00792-f003:**
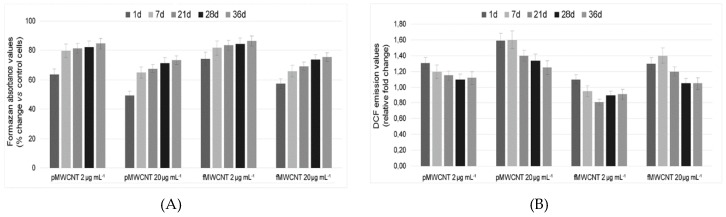
Time course of cellular enzymatic activity and ROS production in MWCNT- treated cells. (**A**) Formazan absorbance during the assayed interval for the assessment of cellular dehydrogenases. The results are expressed as percent change in comparison to control cells. (**B**) DCF emission values during the assayed interval to asses ROS levels. All data are presented as mean ± SEM based on at least three independent experiments.

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
