# Peer review of "Mitochondrial Impairment Induced by Sub-Chronic Exposure to Multi-Walled Carbon Nanotubes"

_ijerph, 2019, doi:10.3390/ijerph16050792_

Round 1

Reviewer 1 Report

The authors made substantial changes in respect to their previously submitted version of the manuscript. These changes considerably helped to improve the manuscript, its comprehension and support their observations.

Some minor language corrections might needed.

Reviewer 2 Report

My previous comments have been addressed. Recommend to accept. 

This manuscript is a resubmission of an earlier submission. The following is a list of the peer review reports and author responses from that submission.

Round 1

Reviewer 1 Report

The authors presented an article about the human exposures to carbon nanotubes (CNTs) and their related health issues and biological interactions such as oxidative effects  The in vitro study results are very interesting and usefull to better know the mitochondrial dysfunction at the sub-chronic exposure with several times of exposure (1, 7, 21, 28 and 36 days) .  Mitochondrial dehydrogenases, pyruvate dehydrogenase kinase 1, cytocrome c release, permeability transition pore opening, transmembrane potential and ROS levels were investigated and measured.

The article is of very good quality, well constructed and organized although the high complexity of the applied experimental methodologies.

All sections sound with the aims and results obtained. Introduction, materials and methods, results and discussion are well described and reproducible.

I not found mistakes or gaps. For all motifs disccussed, I suggest to accept this article in present form.   

Author Response

We thank the reviewer for positive comments on the submitted study

Reviewer 2 Report

-          Pictures (Figure1), show cells after 36 days exposure to nanotubes. It seems from the picture that these cells are all dead. To confirm this point, the use of Trypan blue or/and Hoechst 33342 is suggested. Moreover, based on pictures shown in Fig1, B, E and F the authors assume that nanotubes are internalized in cells. This supposed internalization might be just related to membrane permeability associated also to cell death present in these conditions (Fig 1, B, E, F). All results showing increase of mitochondrial redox potential, pore permeability parameters and so on are meaningless in case cells are dead.

-          MTT is a substrate for mitochondria dehydrogenases but also for any cellular oxidoreductase, specially NAD(P)H oxidoreductases. In case authors want to measure mitochondrial enzymatic activities, they should do it with isolated mitochondria to avoid interference of other cellular enzymes that might reduce MTT. Indeed, the authors’ suggestion of succinate dehydrogenase measurement from the MTT assay is meaningless. They should explain, based on their observations, the reason for having an increase on this activity after cell exposure to nanotubes. This result does not make any sense when the supposed succinate dehydrogenases activity increase in relationship to an increase of Cyt c release from mitochondria. Cyt c release accounts after/during mitochondria depolarization and leads to mitochondrial redox chain uncoupling.  Both Cyt c release and mitochondrial depolarization account in samples after 1 day cell treatment in all tested conditions.

-          There is no reason to assume that cells treated with pMWCNT  2 μg/ml did not suffer cell death.  This type of cell death might not be necrotic but apoptotic that happens delayed in time in respect to the one observed in cells treated with a higher dose of nanotubes (as it seems to be for 20 μg ml-1 of pMWCNT for 1 day, figure 1b). In the line 205, the authors indicate: “the positive controls exposed for 1 day to both MWCNTs, the activity of mitochondrial dehydrogenases was below 60% (P < 0.05) compared to untreated cells. The metabolic inhibition observed at 2 μg mL−1 was not significant and, compared to control cells, dehydrogenase activity was equal to 66.7 and 74.5% for pristine and oxidized nanotubes, respectively” As indicated before, MTT can be used to measure cell death.  Authors should confirm the existence of cell death after 1-day treatment in all samples using other method (i.e.: Trypan blue) as they indicated in Material and Methods.  Moreover, to further assess cell death, they could compliment these measurements with some other apoptosis marker (caspase-3 activation or internucleosomal fragmentation of genomic DNA…)

-          When a mitochondrial protein is analyzed by Western blotting a mitochondrial house keeping protein should be used as a loading control. For example, VDAC/porin or COX should be used instead of B-actin.

-          Results from mitochondrial redox potential and pore permeability measurement after cell incubation for long times are meaningless, in case cells are dead.

-          Formazan does not emit light. The phrase “formazan emission” is wrong.

-          More details should be given in respect to fluorescent measurements with R123 and DCF-DA. Indeed, the intrinsic fluorescence from phenol red present in the RPMI cell media probably interfere with these measurements. The authors should indicate how they avoid this effect.

-          Figures are not in the proper format for publication. The use of a specialized software for graph and diagrams plotting is strongly recommended.

Author Response

-          Pictures (Figure1), show cells after 36 days exposure to nanotubes. It seems from the picture that these cells are all dead. To confirm this point, the use of Trypan blue or/and Hoechst 33342 is suggested. Moreover, based on pictures shown in Fig1, B, E and F the authors assume that nanotubes are internalized in cells. This supposed internalization might be just related to membrane permeability associated also to cell death present in these conditions (Fig 1, B, E, F). All results showing increase of mitochondrial redox potential, pore permeability parameters and so on are meaningless in case cells are dead.

R:We add in the Discussion (lines 301-310) a paragraph reporting also some bibliographical references (Mu et al., 2009, Wang et al., 2016, Trovato et al., 2018) that highlight the existence of different cellular internalisation pathways of nanotubes without perturbing the membrane integrity. These include endocytosis (i.e pinocytosis or, only in specialised cells, phagocytosis) and passive diffusion that are energy dependent and independent mechanisms respectively. Both these mechanisms are due to the high hydrophobicity of the CNTs. 

In the new version, in the Legend of Fig1 we added the percentage of death cells obtained by using Trypan blue. Even if compared to control cells the percentages were higher, the cells were predominantly alive, as confirmed by the proliferation index in the time interval assayed (paragraphs 2.6 and 3.4). In any case we think it is useful to underline that Fig1 shows monolayers treated with the higher concentrations of MWCNTs. This was used as positive controls, to assess the different effects dose-induced. Aimed to evaluate the health effects of sub-chronic exposure to a non-acutely toxic dose, the assayed concentration was 2µg mL-1, on which both the results and discussion are focused. Moreover in the text we added: “The images of cells treated with 2 μg mL−1 are not shown because the nanotube internalisation was barely detectable and the cellular morphology was superimposable to that of the control cells.”

-          MTT is a substrate for mitochondria dehydrogenases but also for any cellular oxidoreductase, specially NAD(P)H oxidoreductases. In case authors want to measure mitochondrial enzymatic activities, they should do it with isolated mitochondria to avoid interference of other cellular enzymes that might reduce MTT. Indeed, the authors’ suggestion of succinate dehydrogenase measurement from the MTT assay is meaningless. They should explain, based on their observations, the reason for having an increase on this activity after cell exposure to nanotubes. This result does not make any sense when the supposed succinate dehydrogenases activity increase in relationship to an increase of Cyt c release from mitochondria. Cyt c release accounts after/during mitochondria depolarization and leads to mitochondrial redox chain uncoupling.  Both Cyt c release and mitochondrial depolarization account in samples after 1 day cell treatment in all tested conditions.

R: We changed in Materials and Methods and in Results the sentences regarding MTT test . In the revised version this test was used to assess cell viability.

-          There is no reason to assume that cells treated with pMWCNT  2 μg/ml did not suffer cell death.  This type of cell death might not be necrotic but apoptotic that happens delayed in time in respect to the one observed in cells treated with a higher dose of nanotubes (as it seems to be for 20 μg ml-1 of pMWCNT for 1 day, figure 1b). In the line 205, the authors indicate: “the positive controls exposed for 1 day to both MWCNTs, the activity of mitochondrial dehydrogenases was below 60% (P < 0.05) compared to untreated cells. The metabolic inhibition observed at 2 μg mL−1 was not significant and, compared to control cells, dehydrogenase activity was equal to 66.7 and 74.5% for pristine and oxidized nanotubes, respectively” As indicated before, MTT can be used to measure cell death.  Authors should confirm the existence of cell death after 1-day treatment in all samples using other method (i.e.: Trypan blue) as they indicated in Material and Methods.  Moreover, to further assess cell death, they could compliment these measurements with some other apoptosis marker (caspase-3 activation or internucleosomal fragmentation of genomic DNA…).

A:As above reported we confirm that cells were predominantly alive even if a low percentage of them went to necrosis or, especially, to apoptosis, as highlighted by the analyses performed in our study. This was observed especially in the short term when still the cells had not implemented homeostatic mechanisms.

-          When a mitochondrial protein is analyzed by Western blotting a mitochondrial house keeping protein should be used as a loading control. For example, VDAC/porin or COX should be used instead of B-actin.

A: We share what was stated by reviewer and to assess a mitochondrial protein in isolate mitochondria VDAC/porin or COX must be used. Thought, our aim was to evaluate the release of cytocrome c from mitochondria in the cytosol. Therefore, as reported in Materials and Methods, “the Cytosolic and mitochondrial fractions were further centrifuged at 11,000 g for 10 minutes to obtain the supernatant containing cytosol fraction.” In the new version we added, “without mitochondria” According this , we used as control a cytoplasmic protein

-          Results from mitochondrial redox potential and pore permeability measurement after cell incubation for long times are meaningless, in case cells are dead.

R: All results confirmed that cells were predominantly alive as also shown by proliferation index. This was, in the assayed MWCNT concentration, only 10% or less lower than the control cells

-          Formazan does not emit light. The phrase “formazan emission” is wrong.

R: We apologize for the mistake, as it was already mentioned in the text we have corrected with "absorbance values”

-          More details should be given in respect to fluorescent measurements with R123 and DCF-DA. Indeed, the intrinsic fluorescence from phenol red present in the RPMI cell media probably interfere with these measurements. The authors should indicate how they avoid this effect.

R: We added two sentences (lines 178-182): “For the analyses, treated cells were washed three times with PBS containing 10 mM D-glucose at pH 7.4. Then in aliquots of cell suspensions (1 x 105 mL-1), prepared by using the same buffer, the emission values were read both before and after the separate addition of the respective probes in order to subtract from the values of the fluorophores those due to a possible auto-florescence.

-          Figures are not in the proper format for publication. The use of a specialized software for graph and diagrams plotting is strongly recommended.

R: For the figures, that were attached in tif format, we followed the jurnal’s guidelines. We believe that the reviewer could only view the manuscript with the figures converted into word format.

Reviewer 3 Report

The manuscript studied the exposure of MWCNTs onto A549 cells to probe their health impacts. The study is comprehensive and recommend to accept after addressing the following comments:

1) It is not common to include "Background", "Methods", and "Conclusions" in the Abstract

2) The results for pMWCNTs and fMWCNTs seems not significant different throughout the tests. The dosage seems to play the dominant roles. If yes, any comments on why the functional groups not delivering significant impacts?

Author Response

The manuscript studied the exposure of MWCNTs onto A549 cells to probe their health impacts. The study is comprehensive and recommend to accept after addressing the following comments:

1)      It is not common to include "Background", "Methods", and "Conclusions" in the Abstract

A: we delete Background, Methods, and Conclusions

2) The results for pMWCNTs and fMWCNTs seems not significant different throughout the tests. The dosage seems to play the dominant roles. If yes, any comments on why the functional groups not delivering significant impacts?

A: In the discussion (lines 385-390) we added a sentence explaining that “The superimposable effects of both CNTs, could be imputable to excessive lengths with regard to the p-MWCNTs (10–20 μm vs. 200–1000 nm) and to higher surface reactivity with regard to the f-MWCNTs (12, 19). Even if the presence of carboxyl groups enhances water dispersibility and causes a reduction in the length to diameter ratio of CNTs, making them more biocompatible, these effects are nullified by the erosion acid-induced in the graphene external layers. This increases the surface reactivity and, consequently, the cellular toxicity (18).

Round 2

Reviewer 2 Report

   Some modifications have been included in the manuscript from the previous version. Noteworthy the authors did not experimentally answer some of my questions that are key to demonstrate their hypothesis.

 - The results obtained using Trypan should be included in the main text rather than in the figure legend. The data reported are preliminary due to lack of standard deviations and n value. Moreover, the authors indicate that the percentages of cell death after 1 day incubation vs 36 days with 20μg/mL of nanoparticles are:  55% vs 36% pMWCNTs and 42% vs 25% fMWCNTs. This is a key experiment that suggest that cells surviving after nanoparticle exposure are able to grow and recover the previous number of cells grown in culture after incubation, as also suggested by the relatively high proliferation indexes vs. controls. Authors should indicate the doubling time of the culture and also count the number of cells before and after treatment to know at the different measured times and correlate and discuss their results with the obtained changes in the number of cells.

- Results from MTT assays are puzzling since authors show “formazan emission value” (it is not emission but absorbance), relative to control cells (non-treated). Therefore, conditions  indicated to do not induced cell death (2ug/mL pMWCNT) show a 60% of MTT reduction (formazan absorbance relative to control cells non-treated, 100%) after 1-day incubation. These results suggest that 40% of cells in these conditions are dead and are contradictory to those values reported with the Trypan blue assay. The authors should revise these values (Trypan blue and MTT assays) and measure cell viability with another dye (i.e.: Hoechst 33342 or another on, as indicated) to fully confirm this point. Moreover, a recovery pattern on MTT assays is observed in cells incubated with 2 or 20ug/ml of nanoparticles for long times (up to 36 days). As indicated before, this could be due to increase of the number of cells rather than associated to changes in the metabolic activity.

- The authors keep indicating the use of MTT to measure mitochondrial dehydrogenase activity as shown in line 212-214: “In the positive controls exposed for 1 day to both MWCNTs, the activity of mitochondrial dehydrogenases was below 60% (P < 0.05) compared to untreated cells” this should be avoided unless measurements are done with purified mitochondrias.

- A 40% percentage of cell death obtained  after 1 day incubation with 2ug/ml pMWCNT correlate with the presence of cytosolic Cyt c as a marker of cell death. In figure 2B, the spots of the sample incubated for 1 and 7 days with 2ug/mL of p or fMWCNT are at least double of those reported for controls, where actin does not seem to change. The figure does not correlate with the table shown below in the same figure (2B). A recalculation of the values reported is suggested.

- Line 297-300 ”Despite the clear tendency to restore the full functionality, our results showed that the impairment MWCNTs-induced of cellular dehydrogenases activities, included succinate dehydrogenase of mitochondrial OXPHOS 2 system, persisted over time.” As previously indicated Succinate dehydrogenase should be measured with specific activity assays and if possible using isolated mitochondrias.